# Fiber/Yarn-Based Triboelectric Nanogenerators (TENGs): Fabrication Strategy, Structure, and Application

**DOI:** 10.3390/s22249716

**Published:** 2022-12-12

**Authors:** Yu Chen, Yali Ling, Rong Yin

**Affiliations:** Wilson College of Textiles, North Carolina State University, Raleigh, NC 27695, USA

**Keywords:** triboelectric nanogenerator, fiber, yarn, fabrication, smart textile, sensor, energy harvesting

## Abstract

With the demand of a sustainable, wearable, environmentally friendly energy source, triboelectric nanogenerators (TENGs) were developed. TENG is a promising method to convert mechanical energy from motion into electrical energy. The combination of textile and TENG successfully enables wearable, self-driving electronics and sensor systems. As the primary unit of textiles, fiber and yarn become the focus of research in designing of textile-TENGs. In this review, we introduced the preparation, structure, and design strategy of fiber/yarn TENGs in recent research. We discussed the structure design and material selection of fiber/yarn TENGs according to the different functions it realizes. The fabrication strategy of fiber/yarn TENGs into textile-TENG are provided. Finally, we summarize the main applications of existing textile TENGs and give forward prospects for their subsequent development.

## 1. Introduction

In the past decade, emerging technologies such as the Internet of Things, artificial intelligence, and big data have developed rapidly [1,2,3]. These technologies continue to change the way humans live and communicate [4,5,6]. At the same time, with the decrease in the size of electronic components and the increase in functionality, people are eager to use more electronic accessories in daily life or wear them directly on the body [7,8,9]. Research in wearables is in full swing. However, the center of the operation of an electronic system is the power supply [10,11,12,13]. The bulky size of traditional batteries, low comfort, and environmental issues limited its application in wearable devices [14,15]. Therefore, many energy harvesting technologies and self-powered systems are studied. Among them, Wang et al. first proposed the triboelectric nanogenerator (TENG) concept in 2012 [16]. It can collect mechanical energy and convert it into electrical energy through a coupled effect of contact electrification and electrostatic induction [17]. Mechanical energy exists in a large amount in our life, and all motion can generate mechanical energy. However, it is also the most overlooked source of energy. The human body is an essential source of mechanical energy, and the human body is also the terminal for the operation of electronic components [17,18,19]. Therefore, integrating the TENG into a wearable system can solve the problems brought by the inconvenience of conventional power sources [20]. In addition, the principle of TENG enables it to be used not only as an energy source but also as a self-powered sensor for sports, health and physical monitoring, signal transmission, and human–computer interaction [13,21,22]. Presently, the miniaturization and multi-functionalities of electronic devices are the research trends [6,18,23]. TENG has shown promising application prospects as a multifunctional self-driving system in the wearable field [24,25,26].

### 1.1. Fundamentals of TENGs

Triboelectric electrification is a phenomenon that has existed for thousands of years without being well studied [27,28,29]. According to several studies, each contact-electrified component acquires a net charge during contact with either a positive or negative polarity rather than a uniform charge distribution [30]. Each surface contains a random “mosaic” of nanoscopic-sized, diametrically opposed regions. They postulated that contact electrification is a multi-step process comprising at least bond cleavage, chemical alterations, and material transfer occurring in discrete patches of nanoscopic dimensions [30]. The model of the triboelectric principle is summarized by Daniel et al. [31]. Based on this research Weon-Guk et al. considered that the charge transfer mechanisms could be categorized into three possible species which are responsible for triboelectricity: electrons, ions, and cleaved bulk materials (Figure 1a) [27]. In the electron transfer model, as two materials come into contact, charges are transferred according to the different charge affinities of the materials [32,33]. Despite the lack of conclusive experimental evidence, this assumption is widely accepted. Moreover, some research proved that transferred electrons are the primary species in contact electrification. The ion transfer model is normally applied in ionic polymers. In this model, mobile ions could transfer from one material to another when the materials are in contact [34,35]. The third model is material Transfer Model [36]. Through Kelvin probe force microscopy, CRS, and XPS analysis, Baytekin et al. found that the contact between the polymers is charged to produce a nano mosaic charge pattern, which is generated by the corresponding material transfer. The nanomaterial fragments transfer the electric charge, creating a triboelectric phenomenon [30].

The TENG was created based on studies on contact electrification. Charge transfer and electrification occur in contact or friction between two different triboelectric materials, resulting in the coupling between triboelectric effects. 

It is well known that TENG has four different working modes: contact-separation mode (CS), Lateral sliding mode, single-electrode mode (SE), and freestanding mode (Figure 1b) [37]. Dielectric materials generate triboelectric charges by contacting each other. Based on the generation of triboelectric charges, an induced potential is generated on the conductor through the relative displacement between dielectric materials, thereby generating electricity [32,34,38]. During the cycle of contact and separation, an alternative current (AC) can be produced. Among them, the lateral sliding mode can be designed to generate direct current (DC). Due to the characteristics of textiles, the vast majority of fiber/yarn-based TENGs were designed based on CS and SE modes. The working principles of these two modes are similar. As shown in Figure 1b, the SE mode includes an electrode and triboelectric material attached. When another triboelectric material with different electron affinity contacts and separates with it, electrons will flow in or out of the electrodes to create an electric current. CS mode has two electrodes that are connected. Each electrode carries a triboelectric material [33]. When the two electrodes come into contact and separate, electrons flow from one electrode to another. Compared with CS mode, SE mode is more straightforward in structure and more flexible in design, so it is widely used in Textile-TENG. CS mode has better power output and stability. Both modes are suitable for working in environments with frequent external forces (such as carpets, insoles, and joint movements) [14].

**Figure 1 sensors-22-09716-f001:**
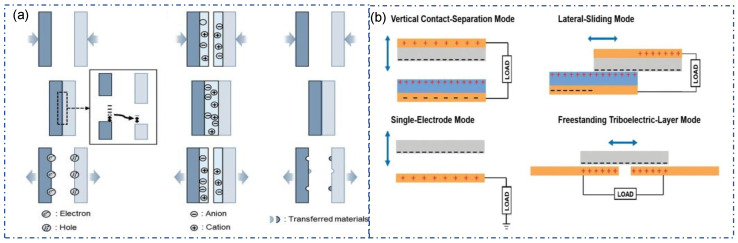
(**a**) Model of electron transfer, ion transfer and material transfer between two insulators. Copyright 2021, American Chemical Society [27]. (**b**) Schematic of four basic TENG working mode: Contact-separation mode, Lateral-sliding mode, Single-electrode mode and Freestanding mode [37]. Copyright 2020, Wiley.

### 1.2. Textile-TENG

Traditional textiles are often used for shade, protection, warmth, and aesthetics. With the development of textile technology, textiles have begun to be given more meaning, and functional textiles have been valued [39,40,41]. Common functional textiles have the functions of water and oil repellency, fire prevention, and UV protection [42]. Nevertheless, these functions are developed based on traditional textiles, and there is no real innovation. With the development of information technology and wearable appliances, a whole new class of functional textiles was born: smart textiles [43,44,45]. As a new type of wear-able system, smart textiles can achieve functions such as energy harvesting, electrical energy generation, human–computer interaction, wearable sensors, and signal output while maintaining the comfort of traditional textiles [46,47].

Portability is one of the issues in developing smart fabrics. Conventional chemical batteries, the most common kind of energy storage, have drawbacks such as size, portability, lifespan, and environmental impact [48,49]. Additionally, standard electrical components have inflexible and rather bulky structures that fall short of meeting wearer demands for comfort [50,51]. In addition, temperature, humidity, sweating, cleaning, friction, and other conditions can have an impact on how well traditional electronic components work [52]. Consequently, these issues have become major challenges in the design of smart textiles.

The daily motion of the human body produces a significant quantity of mechanical energy, which can be captured and transformed into electrical energy via TENG. As an integral part of daily life, textiles are very suitable as carriers for TENG [53,54]. Textile-TENG is a vital part of a self-charging smart textile system that utilizes the triboelectric effect to continuously power the electronic components in the system [55]. In addition, TENG can be designed as wearable electronic components, such as a self-powered system for sensing or a human–machine interface integrated into smart textile systems [56].

There are two main design strategies for Textile-TENG. One is to add triboelectric materials to existing textiles, such as integrating triboelectric materials on conductive fabrics by electrospinning, solution blow spinning or coating [57,58]. However, conductive or triboelectric materials introduced in this way can affect the properties of the original textile. The multi-layer structure also makes the whole system bulk, stiff, and not easy to carry [59,60]. The second way is to design from one-dimensional structures, prepare triboelectric fibers or yarns, and then process them into two-dimensional or three-dimensional structures by weaving [61]. The design method from 1D to 2D gives textile-TENG the freedom of design, which greatly improves the flexibility of the finished product. The textile-TENG designed and woven by f-TENG exhibits good breathability, biocompatibility and deformability [62]. This approach is considered to be the optimal solution for fabricating textile-TENG. At the same time, by investigating the textile TENG’s arrangement, enhancing triboelectric performance, and taking into account the washability and comfort of devices, the triboelectric design based on one-dimensional structure has been realized.

Textile-TENG can be divided into three categories:

1. Fiber-TENG. Fiber is the basic textile unit, a flexible material with a large length ratio to diameter. It can be designed as single fibers or yarns, or it can be mass-produced for further fabrication. Due to its unique working principle, triboelectric fiber/yarn can be considered unique conductive yarn. Compared with ordinary conductive yarn, triboelectric yarn usually has an additional layer of friction material. The majority of fiber-TENG are now constructed as coaxial structures based on insulating synthetic polymer fibers or programmable, large-diameter conductive wires [24]. 

Fiber-TENG usually consists of two parts: conductive material and friction material [24]. The carrier, triboelectric components, and encapsulation layers for triboelectric devices can be made of nonconductive polymer fibers [63]. The electrodes can be made of straight or coiled conductive wires, such as fibers or synthetic conductive fibers [64,65]. As a result, the majority of fiber TENGs are created as coaxial structures using conductive wires that have been wrapped in insulating material. This kind of fiber TENG Usually has a larger diameter and better ductility. Some of them are designed to have multi-layer or multi-core structures to generate triboelectricity through their deformation without further fabrication. Due to the particularity of the structure of fiber TENGs, SE and CS modes are often the primary design strategies. The combination of conductive yarn and triboelectric material can be considered a kind of SE-mode TENG [66]. The triboelectric effect is generated and accompanied by an induced current when outer materials approach and come into contact [67]. When two different triboelectric materials are combined with conductive yarns, and an external circuit connects the conductive yarns, it is CS mode [68]. Electrons will flow in the external circuit when there is a contact and separation between the two conductive fibers [4,69].

2. Yarn-TENG. Yarn-based TENGs are also very common. Such TENGs are usually based on yarn structures, or triboelectricity is generated by structural design after being woven into textiles [25]. Traditional textile materials such as nylon, PET, PU, and silk are also suitable triboelectric materials. Compared to fiber-TENG, yarn-TENG is closer to traditional textile yarns, based on the contact-separation of two or more fiber components.

3. Some other textile-TENGs are mainly based on existing textiles, adding conductive or friction materials to form a multi-layer structure.

So far, the relevant review mainly focuses on TENG’s material selection and processing technology, and few of them mention the structure and design strategy. In this review, we summarized the preparation, structure, and design strategy of one-dimensional triboelectric materials, namely fiber/yarn-TENG. The function of triboelectric yarns often depends on the design of their structure. Therefore, we classify triboelectric yarns based on their structures and working methods. We introduce the materials, preparations, fabrications, and applications of fiber-yarn-TENG. The challenge and outlook of triboelectric fibers/yarns as energy harvesting and sensors was given as well.

## 2. Material Used in Fiber/Yarn Based Triboelectric Nanogenerator

It is generally believed that the strength of triboelectricity depends on the different electron affinities of the two friction materials. The more electron affinity differences between the two materials, the stronger the triboelectricity. The researchers obtained the triboelectric series table by calibrating different materials. TENG should be designed as far as possible by selecting materials with significant differences in electron affinity in the table. As shown in Figure 2, PDMS, PTFE, fluoropolymer, and polyamide are widely used due to their outstanding triboelectric properties [37].

The triboelectric series shows that silicone resin has good triboelectric negativity, easily attracting electrons and being negatively charged [70]. Most worn materials, such as textile fabrics and human skin, are more likely to be positively charged during contact friction. The excellent elastic and mechanical properties of PDMS make it well-suited to be combined with flexible textiles [71]. Therefore, it has become the most commonly used material for preparing triboelectric yarns by a coating method. PDMS has good processability. It is usually formed by the reaction of two components: A-component and B-component silicone. The two-component silicone resin is liquid prior to the reaction, making it easy to deposit on other materials. Polymers that contain fluorine, such as polytetrafluoroethylene (PTFE), fluorinated ethylene propylene (FEP), and polyvinylidene fluoride (PVDF), have been used as electron-negative part of the TENGs owing to the strongest electron attractive ability of fluorine element and the low surface energy [72]. Some common textile materials (such as nylon, polyamide, and polyester) have comparable triboelectric properties [73]. As mature, mass-producible materials, they possess mechanical properties better suited to existing processing and weaving. In addition, it also has properties such as breathability, fashion, and wearability. Compared with the above materials, such materials usually have triboelectricity. Expecting triboelectric charge, the mechanical properties also should be considered carefully. Fluoropolymers have good strength and wear resistance, which are commonly used as insulating layer materials. Fluoropolymers are not stretchable. In addition, it has very strong water repellency, resulting in poor wearing comfort. PDMS has excellent stretchability, good biocompatibility, and no irritation to the skin. However, poor moisture absorption and breathability are challenges in wearable applications. Both relatively good mechanical properties and processability of traditional textile materials make them promising candidates for wearable applications. However, compared with other materials, its triboelectric performance is poor.

## 3. Fabrication Strategy and Structure Design

### 3.1. SE Mode of Fiber/Yarn-TENG

There are mainly two working modes of fiber-TENG: SE mode and CS mode [74]. The difference between the two modes is the number of electrodes in TENG. SE mode has only one electrode and one friction material [75]. This mode of operation requires external friction materials to provide triboelectric charges [76]. CS mode can be understood as two different SE TENGs are connected, and by rubbing each other, the charge flows from one SE TENG to the other SE TENG. The whole system is CS TENG.

According to the structure, there are two main forms of SE fiber-TENG. One is a simple but stable structure that works through contact separation with outer friction material. It usually consists of an inner conductive core and outer triboelectric material. This form of TENG has become the most widely used strategy for fiber-TENG due to its simple structure, diverse preparation methods, and good performance. The commonly used preparation method is coating or wrapping to introduce high-performance triboelectric materials into the electrode. The coating is a common method in textiles, mainly used for post-finishing to coat materials with special functions on textiles. The coating is also the most common method for producing PDMS-based fiber/yarn TENG. Lai et al. directly coated stainless-steel fibers with Ecoflex (Figure 3a) [77]. The resulting triboelectric fiber can be further seed into an elastic textile fabric. The silicone rubber served as the triboelectric material, and the stainless-steel fiber served as the electrodes. A single TENG fiber can output 15 V, and 7 μA when sewn on the fabric. Dong et al. plied and twisted conductive fiber strands as conductive yarn, and then coated PDMS on the outside of the conductive yarn as a triboelectric material (Figure 3b) [78]. As the number of inner conductive strands increases, the electrical conductivity increases and the triboelectric output increases. In addition, the increase in the number of conductive yarns will increase the surface area of the yarn, thus increasing the contact area during friction, and improving the triboelectric performance. For instance, the PDMS-coated triboelectric yarn with a 2 mm diameter and 8-axial yarn electrodes can reach the maximal value of 150 Wm^−1^ at an external stress of 2 G when the applied frequency is 3 Hz. 

Compared with the films formed by coating, nanofibers, as fiber-TENG friction materials, have a larger specific surface area, better air permeability, better comfort, and significant wearability advantages. Electrospinning is a commonly used method for obtaining nanofibers. It can pull polymers from solution into nanofibers and spin them out through an electrostatic field, widely used in textiles, biology, and medicine. Busolo et al. used electrospinning to wrap PVDF nanofibers on carbon nanotube (CNT) yarns (Figure 3d) [79]. It adopts a rotating collection device so that the PVDF nanofibers are wrapped on the CNT yarn to form a core–shell structure. The yarn demonstrated a high wear resistance, withstanding over 1200 rubbing cycles. The washing test showed no damage on the coating, and no decrease in triboelectric performance, proving its high washability and durability. Ma et al. use conductive sliver fiber as core and polyvinylidene fluoride (PVDF) and polyacrylonitrile (PAN) hybrid nanofibers as the shell to make ultralight nano-micro fiber hybrid single-electrode triboelectric yarns (SETY) (Figure 3e) [80]. During the electrospinning process, the authors used two electrospinning devices to spin the nanofiber simultaneously. The conductive yarn was passed through the middle of the metal receiving disc. During the spinning process, the metal disc rotates to pull and orient the hybrid nanofibers to entangle them in conductive silver yarns. This method not only obtains a more stable yarn structure, but also can be designed for scalable production. 

Spinning is another way to warp triboelectric material on conductive yarns. Yu et al. proposed a strategy for TENG textiles by using conductive fibers as the core yarn and traditional textile fibers (cotton, silk, nylon, etc.) as the shell (Figure 3c) [81]. This technique produces soft, flexible, and wearable Triboelectric yarn. Their manufacturing procedures are appropriate for industry. At the same time, different textile materials have different triboelectric properties and electronic affinity and have a certain flexibility in material selection.

**Figure 3 sensors-22-09716-f003:**
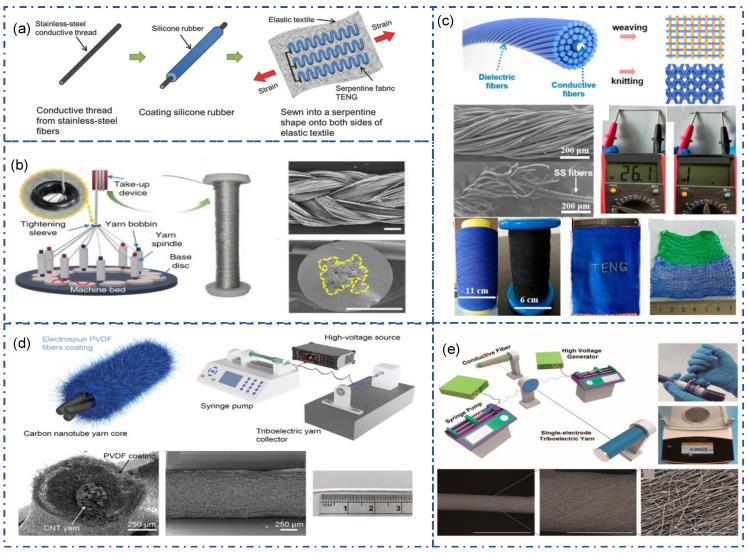
SE mode of fiber/yarn-TENG. (**a**) Fiber-TENG based on silicone coated stainless-steel yarns [77]. Copyright 2017, Wiley. (**b**) Structural design of PDMS-coated energy yarn [78]. Copyright 2020, The Authors, Published by Springer Nature. (**c**) Core–shell-yarn-based triboelectric nanogenerator by Spinning of PU fiber with stainless-steel fibers [81]. Copyright 2017, American Chemical Society. (**d**) Electrospinning of PVDF nanofiber on carbon nanotube yarn [79]. Copyright 2021, American Chemical Society. (**e**) Continuous manufacture of yarn-TENG by Electrospinning of PVDF nano-micro fiber on stainless-steel yarn [80]. Copyright 2020, American Chemical Society.

### 3.2. Stretchable Fiber/Yarn-TENG

As a wearable device for energy harvesting and sensing during exercise, the flexibility and stretchability of Fiber-TENG are critical. There are many options for stretchable triboelectric materials. However, the difficulty of stretchable flexible fiber-TENG is that traditional conductive materials such as metals, graphene, and conductive polymers are not stretchable, limiting the stretching of triboelectric materials. There are three commonly used solutions. The first method is to use “pre-stretch”. This strategy can be used to produce Conductive fibers with ultra-high stretchability. Liu et al. reports this preparation method for the first time (Figure 4a) [82]. Coating by using CNT after stretching the rubber fiber core, followed by a releasing process. When stress is applied to stretch, the length of the elastomer becomes more prolonged, and the diameter decreases. The elastomer returns to its original size when the stress is removed. Driving by the resilience, the CNT layer was pushed back and generated wrinkles. This structure retains electrical conductivity while maintaining extremely high tensile properties, with little change in resistance during subsequent tensile tests.

Similarly, Ning et al. used a three-step method to fabricate fiber-TENGs with high elasticity and ultra-fine wire diameters (Figure 4b) [83]. First, nanosilver particles were attached to pre-stretched spandex core yarn. Carbon nanotube coating was introduced to improve the conductivity. PDMS subsequently encapsulates the composite fiber to form the coaxial structure. This fiber-TENG is ultra-fine with 0.63 mm in diameter and stains up to 140%. It can be folded into different shapes, as shown in Figure 4b.

The second solution is to use stretchable electrodes and stretchable triboelectric materials. Commonly used stretchable electrodes include liquid metals and conductive gels. The introduction of deformable flexible electrodes can maximize the stretchability of PDMS. Yang et al. reported a high-stretchable liquid metal-based TENG (Figure 4d) [84]. This fiber-TENG used liquid metal Galinstan as the stretchable electrode and PDMS as the outer material. First, a twisted iron wire on which the releasing agent had been evenly sprinkled was covered with the silicone rubber mixture. Peeling off the silicone rubber from the wire after naturally solidified, cured fiber-shaped silicone rubber with a hollow structure was obtained. The two ends of the device were then sealed after acquiring the hollow silicone rubber in the form of a fiber, and the liquid metal was then injected into the hollow area. Due to its superior mechanical performance and liquid metal electrodes, this TENG demonstrated a high strain as PDMS (300%). The V_oc_, I_sc_, Q_sc_, and average power density of the 6 × 3 cm^2^ LM-TENG are 354.5 V, 15.6 μA, 123.2 nC, and 8.43 mW/m^2^, respectively.

In addition to liquid metal, conductive gel is another option for a stretchable electrode. Jing et al. demonstrated a fiber-TENG with organogel as electrode (Figure 4c) [85]. The 4-acryloylmorpholine (ACMO) monomer, propylene carbonate solvent, and pre-cured liquid were photo-crosslinked to create the gel electrode in a clear, thin silicone hollow fiber. The tensile test revealed that even after being stretched to almost 200% of its original length, the inner electrode core of the GS-fiber kept its conductivity while the core/shell structure of the fiber was still preserved. Dong et al. reported a stretchable fiber-TENG based on conductive hydrogels (Figure 4e) [86]. This strong stretchable gel-electrode-based triboelectric fiber (GETF) can withstand tensile forces over 1.2 mpa and maintain conductivity at 600% deformation. The salting effect of NaCl in water leads to a strong interaction between PVA and glycerin, which can significantly improve the gel’s mechanical properties, enabling it to work at −20 °C without freezing.

**Figure 4 sensors-22-09716-f004:**
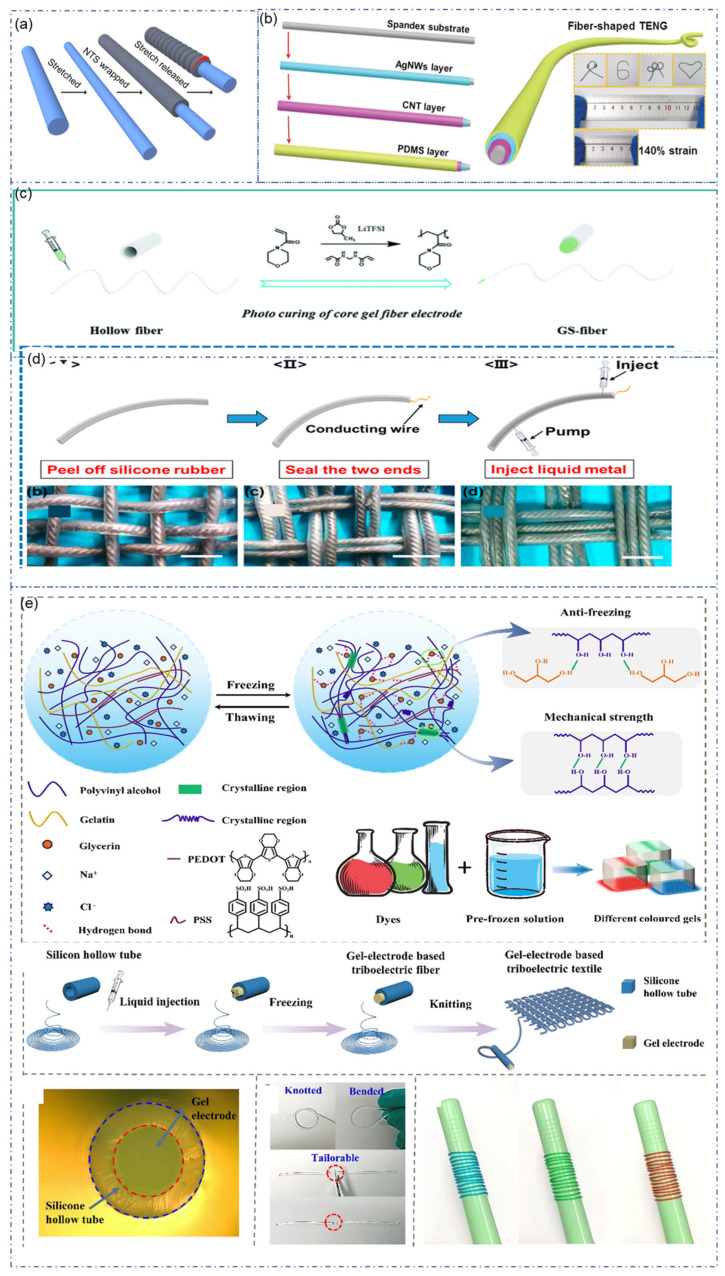
Strategy of stretchable fiber/yarn-TENG. (**a**) Pre-stretching for production of superplastic sheath-core fiber [82]. (**b**) Superplastic fiber-TENG by coating of AgNWs, CNT and PDMS on spandex fiber [83]. Copyright 2021, Wiley. (**c**) Gel electrode-based fiber-TENG by injection of polymer solution into hollow fiber [85]. Copyright 2021, Royal Society of Chemistry (**d**) Stretchable fiber-TENG with liquid metal as electrode [84]. Copyright 2018, American Chemical Society. (**e**) High strength fiber-TENG based on conductive hydrogel [86]. Copyright 2022, Springer.

The third method introduces a coil structure to render non-stretchable conductive material a certain stretchability, like a spring. Park et al. demonstrated a stretchable triboelectric fiber using a sandwich structure with silicon-conductive yarns-silicon (Figure 5a) [87]. The conductive yarns were introduced by convolving around silicon rubber fiber. Then, another silicon rubber was coated on it as a shell for contacting. Gong et al. reported a continuous and scalable method for super-stretchable triboelectric yarn [88]. As shown in Figure 5b, stainless-steel yarn served as the conductive material. PDMS was melted and introduced with frozen compressed gas. Pretension was applied to the pulleys. Due to the difference in the rotational speed of the compaction roll and the take-up roll, the pre-stretched PDMS is released, driving the inner stainless-steel yarn into a helix structure. When the obtained yarn is stretched, the contact surface between the core yarn and the outer layer material decreases while the surface area of the outer layer material increases, thereby creating a potential difference. Alternating current can be produced via repeatedly stretching and releasing the SETEY.

### 3.3. Self-Trigger Fiber/Yarn-TENG

However, outer friction materials significantly limit TENGs with this type of structure, and the output performance and signal are unstable. For example, when the outcome triboelectric material has a similar electron affinity with the TENG fiber, it will weaken the output performance. Therefore, a strategy for “self-trigger” SE fiber-TENG was investigated. This type of SE TENG has more than one triboelectric material. By introducing “gap” between internal triboelectric materials, the TENG can achieve triboelectric effect through contact of each triboelectric material under deformation or pressure, without considering the influence of external materials. Gao et al. reported a core–spun coating yarn with hierarchical structure that utilize nylon and PDMS as triboelectric materials (Figure 5c) [89]. First, cotton fiber was convolved on silver-plated nylon as core fiber, followed by nylon coating. The obtained fiber was braided into flexible thread and coated with PVA. The exact process was applied with PDMS coating replaced by nylon coating. Then, the PDMS-coated yarn was braided on nylon-coated yarn with PVA, forming a sandwich structure. Finally, the PVA was dissolved and generated a “gap” between PDMS and nylon. Under deformation, a triboelectric effect occurs on PDMS and nylon yarns. Pre-stretching is another way to achieve a “gap” between materials. An abrasion-resistant and waterproof stretchable fermat-spiral-based energy yarns (FSBEY) was reported (Figure 5d) [90]. Spandex yarn was used as an elastic core and pre-stretched with braided conductive yarns. Then, this thread was further pre-stretched and warped with nano PVDF-TrFE fibers. During stress release, the nanofiber layer springs back and forms wrinkles, resulting in a small gap between conductive fibers and PVDF-TrFE nanofibers. In this case, conductive fiber served as both electrode and friction material. By stretching, this FSBEY can generate 5 V as V_oc_.

### 3.4. CS Mode Fiber/Yarn-TENG

Although SE fiber-TENG has many advantages in design, it has problems such as small output and unstable signal. On this basis, CE fiber-TENG was introduced. Generally, there are two different forms of CS mode fiber/yarn TENG. The first one can be obtained by combining two different SE fiber-TENG. Ye et al. reported ultrastable and high-performance silk energy harvesting textiles (EHT) using CS mode TENG (Figure 6a) [91]. This EHT contains two triboelectric yarns: Silk fiber warped stainless steel yarn and PTFE fiber warped stainless steel yarn. After the fabrication, two fabrics are connected and generate triboelectric output by contact-separation of each other. Generally, CS mode TENGs are realized by the contact separation of two different TENGs without the need for outer materials. Interestingly, Guan et al. used another approach to achieve CS mode (Figure 6b) [92]. In their design, electrospun nylon and PVDF-TrFE nanofibers were wrapped around stainless-steel threads. Subsequently, these two TENG yarns were woven together as warp and weft. When there are external materials in contact, there is no contact separation process between the two yarns due to the tight woven structure. Since these two materials have strong but different triboelectric properties, outer materials with any type of electron affinity can induce a triboelectric effect, causing electrons to flow from one pole to the other. 

The second method integrates multiple triboelectric materials and electrodes on a single fiber. This type of CE fiber-TENG has a coaxial or core–shell structure. A gap is introduced between two different triboelectric materials to form contact to generate triboelectric charges, then connect the outer electrode to the inner electrode to form a circuit. Yu et al. demonstrated a complex coaxial triboelectric nanogenerator fiber (CTNF) with high performance [93]. As shown in Figure 6c, CTNF has a total of seven layers. First, PDMS fiber was applied as core fiber. Then, the CNT was coated on PDMS fiber as an inner conductive layer. PMMA microspheres and PDMS were deposited as triboelectric material. Sucrose particles were coated on PMMA as a sacrificial layer which will dissolve to generate an air gap between PMMA and PDMS layers. Another layer of CNT was coated on PDMS as an outer electrode. Finally, PDMS was deposited on top for protection of the conductive layer. All materials were coated on pre-stretched PDMS core fiber to obtain stretch properties. The dissolved sucrose created a ~10 μm space for contacting and separating the PDMS and PMMA layer, generating ~2 V as V_oc_ and 200 nA as I_sc_ when the length of CTNF is 2.5 cm, and the pressing force is 10 N. He et al. utilized copper wire convolved with silicone rubber to produce highly stretchable CS-TENG. Conductive ink made of CNT and polymer was coated on silicone rubber as an inner electrode, followed by another coating layer of silicone rubber. Copper microwire was convolved on the fiber as the outer electrode as well as the outer friction material. The copper coil will contact and separate with silicone rubber during stretching.

### 3.5. Scalable Production of Fiber/Yarn-TENG

Due to the limitation of contact area, single fiber-TENG is challenging to realize the purpose of TENG as an energy source. Fabrication of fiber-TENG can not only improve the surface area and triboelectric performance, but also maintain good wearability. However, the large-scale production of triboelectric yarns is the core issue. The complex structure, various materials, and different processing of triboelectric yarn are the main problem when considering large-scale production. A few strategies for scalable triboelectric yarns are listed. As mentioned above, spinning has been proven as a robust method for scalable production of fiber/yarn-TENG. Traditional textile materials have excellent properties and are easy to apply. At the same time, fiber/yarn TENG produced by spinning can be better used for subsequent fabrication. They can be directly used in existing textile machines, such as weaving, knitting, sewing, and other equipment. Braiding is a good candidate for the fabrication of fiber/yarn-TENG. It allowed triboelectric fiber to be directly warped on a conductive thread to make yarn-TENG. As mentioned in the last section, the production or coating of a hollow silicone tube with the injection of conductive material can also be designed as scalable production. A modified electrospinning method was reported for scalable production of nanofibrous yarn for wearable smart textile (Figure 7) [94]. It is a one-step method based on electrospinning which involves a pair of positive and negative power supplies. The prepared polymer solution was put in different nozzles connected to the positive and negative power supplies. Under the action of electrostatic force, nanofibers are spun out and form a spinning triangle above the drum. The collector was rotated, and the spun fibers were twisted in the spinning triangle around the metal core.

## 4. Fabrication of Fiber/Yarn TENG

Weaving is the most common method of textile weaving. Weaving fabric with simple structure, stability, and excellent performance is also prevalent in preparing TENGs. In addition to Fiber-TENG, some films and strip TENGs are also woven in this way. The structure of warp and weft in weaving has significant advantages in fabricating CS-TENG with dual electrodes. The most direct method is to weave fiber-TENG as cloth. However, the contact material properties affect the performance of this SE-mode TENG. A better way is by two or more triboelectric yarns weaved together and working in CS-mode. For example, a textile-TENG made of nylon and polyester fabric strips with silver fiber fabric can be weaved together and generate a I_SC_ up to 1 μA and V_oc_ about 90 V via free-standing mode (Figure 8a) [88]. The weaving fabric consists of single-electrode triboelectric yarn (SETEY) (weft yarn) and MPAN-stainless steel yarn (warp yarn) is demonstrated in Figure 8b. This fabric is highly stretchable and can generate V_oc_ of 10 V to 35 V, depending on circuit connection patterns. As shown in the figure, the weft-connection single-electrode pattern and warp-weft-connection single-electrode pattern were considered optimal output circuits.

Knitting is another common way of weaving. Compared with woven fabrics, knitted fabrics are softer and have good elasticity and extensibility. It is also better in breathability and comfort than other fabrication strategies, which are more suitable as a smart textile. At the same time, the flexible coil structure design of knitting gives Textile-TENG a larger design space. Benefiting from the interaction between the coils, some knitted TENGs can achieve the effects of energy harvesting and signal output through stretching and deformation without external material contact. Chen et al. reported a 3D double faced interlock fabric TENG (3DFIF-TENG) for motion energy harvesting and self-powered wearable sensor (Figure 8d) [95]. Three-ply cotton yarn and PA composited yarn with silver and silicone coating were used as yarn-TENG for knitting of interlock fabric. Two working modes were achieved. The first one is the SE mode that is triggered with external triboelectric material. The second one is the contact and separation between cotton yarn and PA composite yarn during the stretching or deformation of the fabric. This novel structure renders it high sensitivity and gives it potential as a pressure sensor and tactile sensor. Inspired by the overlapping structure of organisms, Niu et al. fabric for outdoor rescue and human protection (Figure 8c) [96]. They utilized three types of yarn: PTFE yarn, nylon yarn, and Ag-plated nylon yarn. BSK-TENG has a three-layer structure and is woven by a high-speed V-bed flat knitting machine at one time. The upper layer is made of PTFE yarn to form a scale-like dielectric layer. The middle and bottom layers are plain weave structures woven with nylon and conductive yarns used as friction materials and electrodes, respectively. BSK-TENG has no vertical spacer layers, retaining the flexibility and wearability of knitting fabric. Simply pressing BSK-TENG, this self-powered sensor sends wireless signals to the App in the smartphone via Bluetooth low-energy (BLE). 

Besides fabric, other textile methodologies such as sewing, embroidery, and braiding can also be used for textile-TENG. Dong et al. demonstrated a four-step rectangular braiding process was used to create a three-dimensional five-directional braided (3DB) structure combining energy yarn coated with PDMS as the braided yarn and eight-axial winding yarn as the axial yarn. (Figure 8f) [78]. The connecting line from the middle of the yarn carriers will be the braided yarn trace in the cross-section. After a four-step braiding cycle, the axial yarn carrier only passes in the X direction before returning to the initial pattern. Yu et al. reported an ultra-fine and durable copper thread used in embroidery for energy harvesting and sensing (Figure 8e) [97]. A 5-ply ultra-thin copper thread (150 μm as diameter) was produced by introducing a high twist for five polyurethane-coated copper wires (50 μm in diameter). This plied structure renders it high strength while maintaining good flexibility. It can be directly embroidered to different patterns with a commercial embroidery machine. The embroidery fabric can be placed in a different position on humans for energy harvesting and motion capturing.

**Figure 8 sensors-22-09716-f008:**
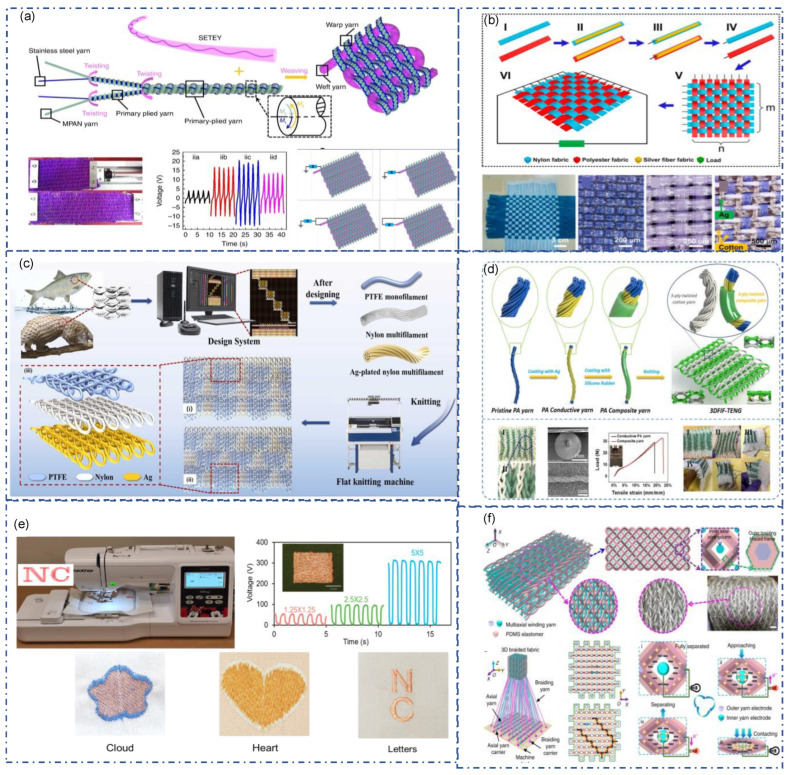
Fabrication of fiber/yarn-TENG into textile-TENG. (**a**) Stretchable weaving fabric made of single-electrode triboelectric yarn as weft yarn and stainless-steel/MPAN plied yarn as warp yarn [88]. Copyright 2019, The Authors, Published by Springer Nature. (**b**) Textile-TENG by weaving of nylon/silver conductive strip and polyester/silver conductive strip [98]. Copyright 2014, American Chemical Society. (**c**) A bionic scale knitting TENG made of PTFE, nylon and Ag-plated nylon monofilaments [96]. Copyright 2022, Elsevier. (**d**) 3D braided triboelectric nanogenerators made of PA composite yarns [95]. Copyright 2020, Elsevier. (**e**) Embroidery TENG made of ultra-fine and high twisted enameled copper wire [97]. Copyright 2022, Elsevier. (**f**) Textile-TENG based on four step rectangle braiding technique [78]. Copyright 2020, The Authors, Published by Springer Nature.

## 5. Application

### 5.1. Energy Harvesting

Many studies have shown that TENG is a promising energy harvesting solution. The output of some fiber/yarn based TENGs are summarized in Table 1. Simply by flapping, it can quickly obtain an output voltage of up to several hundred volts and can be directly used for LED lighting (Figure 9c) [95]. When connected with a rectifier circuit, textile-TENG can power capacitors or other small electronics such as a pedometer, mini calculator, and electronic watch (Figure 9a) [84]. Some research transmitted the energy from flowing water (rain) to triboelectric output, which can be used to power LED screens (Figure 9b) [90]. However, it is difficult to use it directly to power electronic components due to its unstable performance and usually AC output. A better solution is to combine TENGs with batteries or capacitors to form self-charging systems. Dong et al. demonstrated a fiber-TENG that combines self-charging and energy storage systems (Supercapacitor) on the same yarn (Figure 9d) [99]. This fiber uses H_3_PO_4_/poly(vinyl alcohol) (PVA) electrolyte as the inner core and supercapacitor, and carbon nanotubes as electrodes connected to the supercapacitor. PDMS provides strength and flexibility to the yarn while acting as a triboelectric material. In addition, PDMS also separates SC and TENG, avoiding mutual interference. The performance of fabricated TENG: a V_oc_ of 42.9 V, Q_tr_ of 15.1 nC, maximum I_sc_ of 0.51 μA, and maximum output power of 1.12 μW.

### 5.2. Sensors

Unlike traditional sensors, TENG can output signals without an external power supply, rendering its high potential as a self-power sensor. The signal output of TENG is affected by force, frequency, deformation, and environmental factors and can be used for healthcare monitoring, human motion detection, and human–machine interaction. Both fiber-TENG and fabric-TENG can be used in sensing system design. The properties of SE mode TENG influenced by foreign materials make it worthwhile for material identification. For example, the triboelectric affinity of different textiles is different, so Textile-TENG can “recognize” it. As shown in Figure 10a, when contact with different fabrics such as cotton, silk, polyester, and nylon, the open circuit voltage produced by Textile-TENG is significantly different [80]. The high sensitivity of applying force renders it great potential as a stress sensor. As shown in Figure 10b, a weighting cushion sensor was demonstrated, with high accuracy for measuring the occupant’s weight [95]. In addition, a fully textile stress sensor successfully detects the weight of liquid in the bottom (Figure 10c) [95]. By connecting each yarn-TENG individually, a sensing fabric was prepared. It can track or record the action of the finger or external force at each point of the fabric (Figure 10d) [80]. 

Gesture detection is the most common application of fiber-TENG. By sewing with gloves, the fiber-TENG deforms according to the finger’s movement, generating the triboelectric signal (Figure 11a) [100]. Different gestures are distinguished based on signal strength, frequency, and pattern. By placing textile-TENG in joints or insoles, the electrical signal generated by movement can be directly applied for analysis without needing a power source (Figure 11c) [97]. Human–computer interaction is also a prominent application of TENG. A simple, self-powered, fully textile keyboard/controller is designed (Figure 11b) [100]. No power supply is required, and this interactive device can be directly controlled by a mobile phone or laptop, for music control or phone call.

The real-time monitoring of human health has attracted much attention. It has high requirements for wearing comfort, power limitation, and signal transmission, which are the advantages of Textile-TENG. Zhao et al. integrated biological enzymes on fiber-TENG to obtain a biosensor for real-time sweat and motion analysis (Figure 11f) [101]. Glucose, creatinine, and lactate acid in sweat can be detected by the coupling effect of triboelectric and enzymatic reaction (surface-triboelectric coupling effect). The coil structure of varnished wires with polyaniline (PANI) stretchable conductive yarns generates triboelectric signals during motion. A wireless, self-powered health monitoring system was reported by Meng et al. (Figure 11e) [102]. A flower shape of textile-TENG composed of polyester-metal hybrid yarns and Ag-coated fabric was placed on the wrist. The deformation induced by blood pressure generating electricity is attributed to a conjunction of triboelectrification and electrostatic induction. The signal can be transferred to a cell phone for continuous monitoring for human obstructive sleep apnea-hypopnea syndrome diagnosis with filtering and amplification. Pulse wave and respiratory wave measurements were achieved through a sensor array based on a textile-TENG with high sensitivity for subtle pressure detection (Figure 11d) [103]. This textile-TENG was knitted with fiber-TENG and nylon yarns in a cardigan stitch. Two sensor arrays placed in the abdomen and wrist can obtain up to 7.84 mV/pa for sensitivity and 20 ms as response time. A wireless mobile system was designed to transmit a signal from the sensor array to the App in the cell phone.

**Figure 11 sensors-22-09716-f011:**
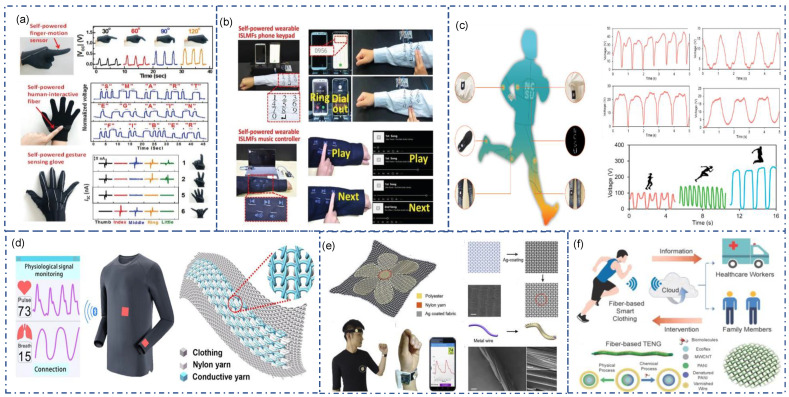
Multifunctional sensor based on textile-TENG (**a**) Self-powered gesture and finger motion detection via fiber-TENG [100]. Copyright 2021, Wiley. (**b**) Self-powered fully textile based smart controller/keyboard via fiber-TENG [100]. Copyright 2021, Wiley. (**c**) Motion detection in different position of human body by embroidered fiber-TENG [97]. Copyright 2022, Elsevier. (**d**) A smart t-shirt with sensor array based on a textile-TENG for subtle pressure detection [103]. Copyright 2020, The authors, Published by American Association for the Advancement of Science (**e**) A fully textile wristband for heart rate and sleep quality monitoring [102]. Copyright 2019, Elsevier. (**f**) Wearable and biocompatible fiber-TENG for real-time sweat analysis and body motion capturing [101]. Copyright 2022, Elsevier.

## 6. Challenges and Summary

For fiber/yarn TENG, simple and reliable structure, flexible design, and size selection are its most significant advantages. According to the triboelectric series, the material selection is relatively fixed, mainly PDMS, PTFE, nylon, and other materials. These materials have their own advantages and disadvantages. For example, PDMS and fluoropolymers are extremely triboelectric, but poor in terms of mechanical properties and comfort. Textile materials such as nylon are excellent in comfort and processability, but most of them are tribo-positive materials with poor triboelectric performance. Therefore, there are often trade-offs in the selection of materials. The application of new technologies such as electrospinning of fluoropolymers, which are produced in the form of nanofibers and applied in fiber/yarn TENG. These methods greatly improve the wearing comfort of polymers. The addition of conductive fillers can further improve the triboelectric properties of these materials [37]. For wearable applications, there are fewer choices of existing materials. The synthesis and application of new polymers are expected.

Through structural design and different processing methods, fiber/yarn-TENG can possess the advantages of textiles, such as high elasticity, breathability and durability, and be used in fields such as energy harvesting, sensor design, and human–computer interaction. It is the popular direction of TENG at present. However, the contact surface area’s size limited its energy harvesting application. The current common method is fabrication, converting the yarn into cloth. For use as a sensor, the susceptible inherent of triboelectricity is a hindrance. According to the triboelectric model, the triboelectric output depends on the exchange of electrons when the material is rubbed/contacted. This process depends on the external environment, such as humidity, temperature, and other influences. The change of environment will change the triboelectric performance, resulting in instability as a sensor. Meanwhile, the accumulation of surface charges on triboelectric materials is the key to triboelectric output. The generation of triboelectric charges is often accompanied by rapid loss, so it is easy to change with time. A possible solution is to introduce special microstructures on the surface of triboelectric materials, such as superhydrophobic structures, to increase the stability of the triboelectric properties of materials and avoid the influence of environmental factors. Sealing of triboelectric devices is also a possible solution.

Thanks to the convenient fabrication method, fiber/yarn-TENG can be directly processed into textile-TENG, demonstrating great potential in smart wearable, wearable sensor, self-powered system, health monitor and human machine interface, which greatly expands applications of fiber/yarn TENG. In addition, textile-TENG significantly improves the complex multi-layer structure and poor wearing comfort of traditional composite TENG.

Overall, textile-TENG still faces some challenges: First, mass manufacturing is still tricky. Several studies have achieved large-scale production of triboelectric yarns and textile-TENGs. However, most of these are still lab scales, which greatly limit production efficiency. Second, the function is single. At present, textile-TENG is mainly used as an energy harvesting and sensor. As a functional conductive textile, it has more potential applications, such as water repellency, fire protection, and thermal regulation. Third, the internal resistance of textile-TENG as an electrical energy source is too large, and the actual electrical output is smaller than the output voltage. As a sensor, its performance is affected by many factors, and its stability is poor. Triboelectric sensors still require much research in terms of precision and accuracy before they can be put into practical use. As a self-powered sensor, it has broad application prospects in the wearable field.

## Figures and Tables

**Figure 2 sensors-22-09716-f002:**
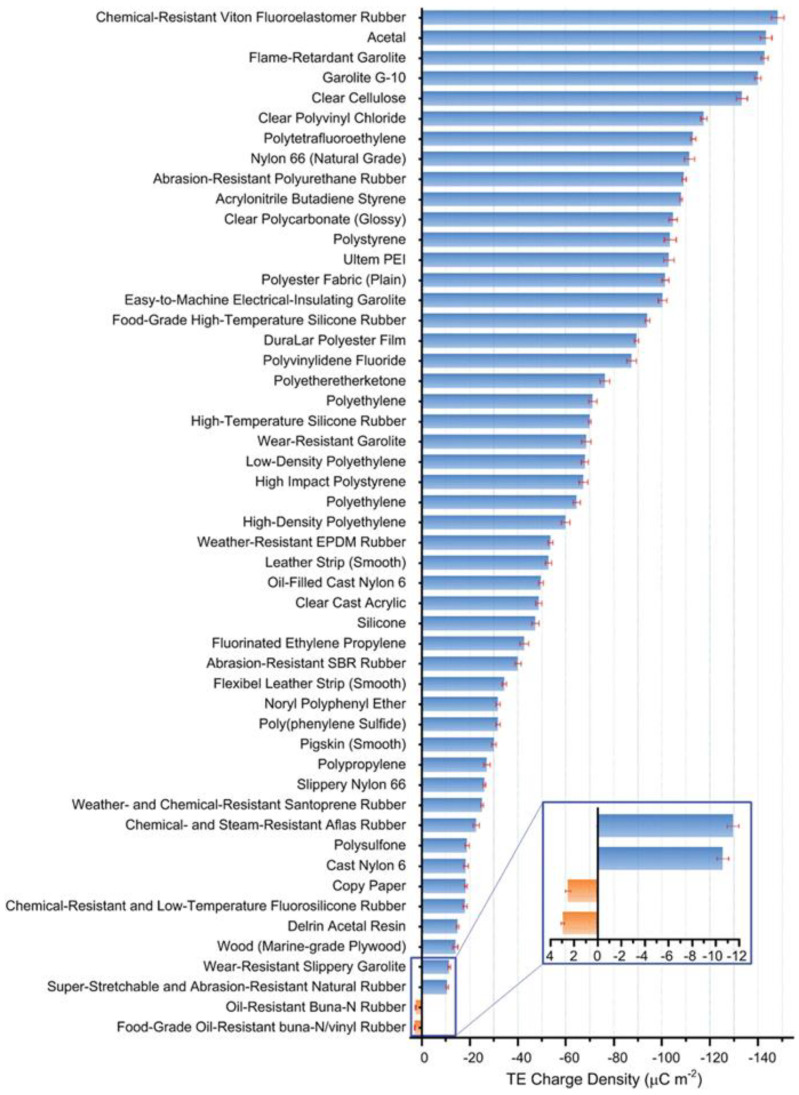
Triboelectric serious of common materials. Measured by contacting with mercury [37]. Copyright 2019, Wiley.

**Figure 5 sensors-22-09716-f005:**
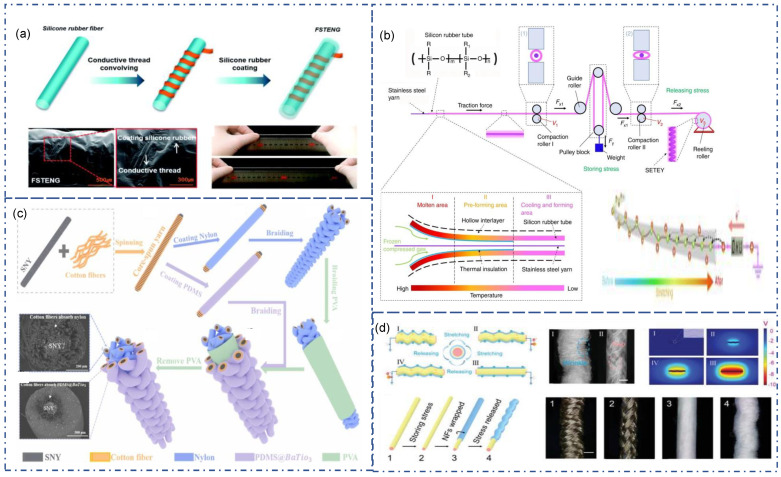
(**a**) Elastic fiber-TENG prepared by coating of PDMS on copper wire winded silicone rubber core [87]. Copyright 2017, Royal Society of Chemistry. (**b**) A scalable, stretchable fiber-TENG fabrication method via external tension to form a coil structure of conductive fiber [88]. Copyright 2019, The Authors, Published by Springer Nature. (**c**) Self-trigger fiber-TENG designed by gap formation via soluble PVA [89]. Copyright 2021, Elsevier. (**d**) Preparation strategy of stretchable fermat-spiral-based energy yarn [90]. Copyright 2021, Wiley.

**Figure 6 sensors-22-09716-f006:**
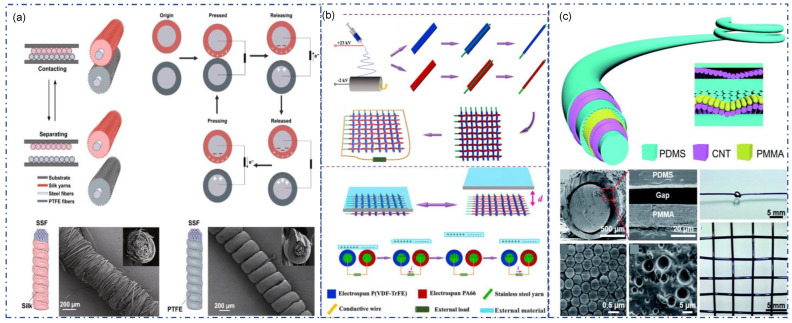
Strategy of CS-mode fiber/yarn-TENG. (**a**) CS-mode of yarn-TENG by silk/PTFE wrapped on stainless-steel fibers [91]. Copyright 2020, Springer. (**b**) CS-mode yarn-TENG by electrospinning of PVDF-TrFE and PA66 on warped on stainless-steel fibers [92]. Copyright 2021, Elsevier. (**c**) CS-mode fiber-TENG prepared by coating of CNT/PMMA/PDMS on silicone rubber tube, with sucrose particles as “gap“ [93]. Copyright 2017, Royal Society of Chemistry.

**Figure 7 sensors-22-09716-f007:**
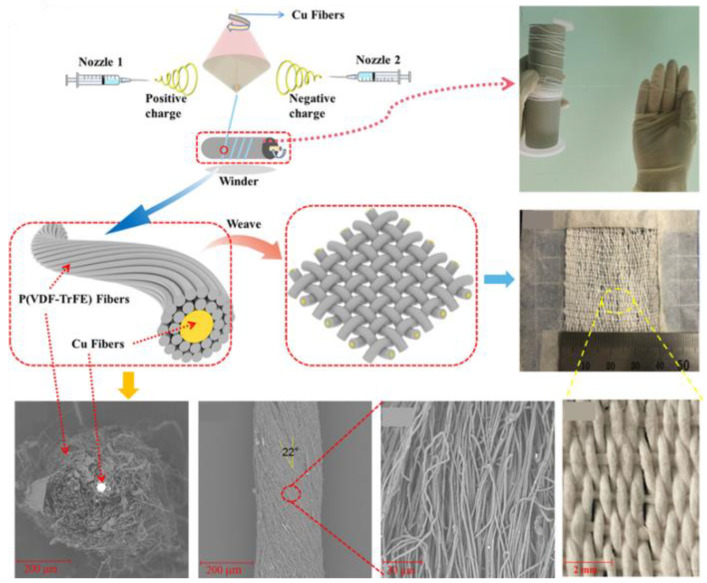
Scalable production of nano-micro fiber/yarn-TENG by one-step electrospinning technique [94]. Copyright 2021, American Chemical Society.

**Figure 9 sensors-22-09716-f009:**
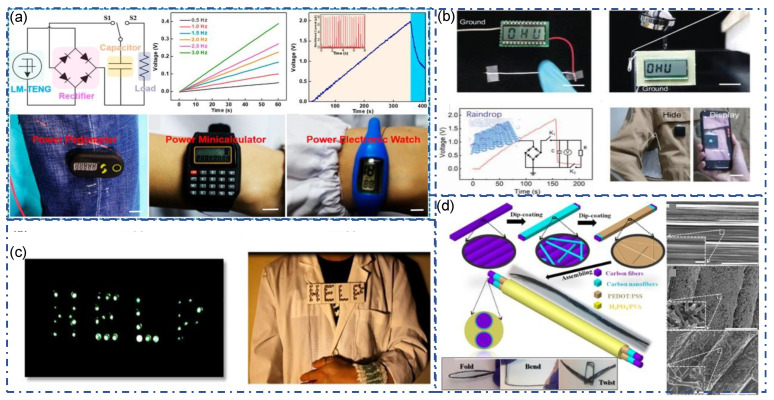
Fiber/yarn-TENG as energy source. (**a**) Demonstration of TENG as an energy source to drive small electronic devices [84]. Copyright 2018, American Chemical Society. (**b**) fiber/textile-TENG driven by mechanical extrusion or water flow, lighting up LED screen [90]. Copyright 2021, Wiley. (**c**) Textile-TENG as power source directly light up LEDs [95]. Copyright 2020, Elsevier. (**d**) A fiber-TENG that combining self-charging systems and energy storage systems (Supercapacitor). Copyright 2017, American Chemical Soc.iety.

**Figure 10 sensors-22-09716-f010:**
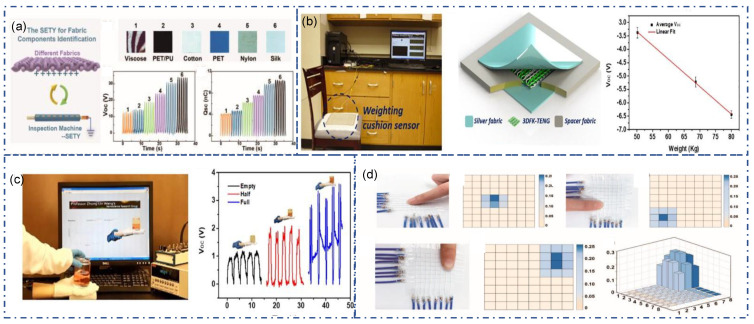
Textile-TENG as sensor. (**a**) The single electrode TENG yarn for fabric components identification [80]. Copyright 2020, American Chemical Society. (**b**) A weighting cushion sensor made of three-dimensional five-directional braided textile-TENG [95]. Copyright 2020, Elsevier. (**c**) Stress sensor made of three-dimensional five-directional braided textile-TENG [95]. Copyright 2020, Elsevier. (**d**) Smart fabric for biomechanical sensing [80]. Copyright 2020, American Chemical Society.

**Table 1 sensors-22-09716-t001:** Output of fiber/yarn based TENG.

Refs	Work Mode	Power Density	Triboelectric Material
[78]	CS	26 W m^−3^	PDMS/Ag-coated yarn
[80]	SE	336.2 μW/m	PAN/PVDF nanofibers/Acrylic
[81]	SE	60 mW m^–2^	Spandex/Polyester
[84]	SE	8.43 mW/m^2^	Skin/PDMS
[88]	CS	12.5 μW/m	Polyacrylonitrile/PDMS
[90]	SE	1.25 W/m^2^	Nylon/PVDF
[91]	CS	3.5 μW/m^2^	Silk/PTFE
[92]	CS	93 mW/m^2^	PA66/PVDF-TrFE/Rubber
[97]	SE	245 μW/m	PU/PTFE

## Data Availability

Not applicable.

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
