# Peer review of "Fiber/Yarn-Based Triboelectric Nanogenerators (TENGs): Fabrication Strategy, Structure, and Application"

_sensors, 2022, doi:10.3390/s22249716_

Round 1

Reviewer 1 Report

The authors made a comprehensive review of the fiber/yarn triboelectric nanogenerators, including working principles, materials, structures, preparation strategy, and potential applications. The outcomes of this work are interesting and useful for the related researchers. Yet more comparisons among different textile structures like fibers or yarns and the working mode of textile nanogenerators are suggested to be added in the manuscript. Moreover, more discussion should be added on the challenge of fiber/yarn triboelectric nanogenerators. Therefore, this manuscript is suggested to be published if the above issues are addressed. More specific comments are listed below.

1.     Comprehensive consideration should be made when we select the materials for fiber/yarn-based triboelectric nanogenerators. For example, expecting triboelectric charge, the mechanical properties also should be considered carefully.

2.     There are several modes of fiber/yarn TENG. The authors have summarized them but did not make enough discussion on the selection of modes. It is suggested to add more discussion on this.

3.     In the application part of energy harvesting, there is no discussion on the outputs of these fiber/yarn TENG. The authors are suggested to add some discussion on the electric outputs.

4.     More discussions on the challenge and potential solutions of fiber/yarn TENG for real applications are added in the section on challenges and summary.  

5.     On page 3, lines 107-109, the authors state that “temperature, humidity, sweating, cleaning, friction, and other conditions can have an impact on how well traditional electronic components work. Consequently, it is important to produce textiles that are self-sufficient.”. The authors are suggested to make clearer discussion.

Author Response

Thank you very much for your attention and comments on our paper. We have revised the manuscript carefully according to your kind advice and suggestion. Below please find the following responses and actions to the reviewers’ comments.

The authors made a comprehensive review of the fiber/yarn triboelectric nanogenerators, including working principles, materials, structures, preparation strategy, and potential applications. The outcomes of this work are interesting and useful for the related researchers. Yet more comparisons among different textile structures like fibers or yarns and the working mode of textile nanogenerators are suggested to be added in the manuscript. Moreover, more discussion should be added on the challenge of fiber/yarn triboelectric nanogenerators. Therefore, this manuscript is suggested to be published if the above issues are addressed. More specific comments are listed below.

  1. Comprehensive consideration should be made when we select the materials for fiber/yarn-based triboelectric nanogenerators. For example, expecting triboelectric charge, the mechanical properties also should be considered carefully.

 A: Thank you for your suggestion. We have added more discussion in this part.

Expecting triboelectric charge, the mechanical properties also should be considered carefully. Fluoropolymers have good strength and wear resistance, which are commonly used as insulating layer materials. Fluoropolymers are not stretchable. In addition, it has very strong water repellency, resulting in poor wearing comfort. PDMS has excellent stretchability, good biocompatibility, and no irritation to the skin. However, poor moisture absorption and breathability are challenges in wearable applications. Both relatively good mechanical properties and processability of traditional textile materials make them promising candidates for wearable applications. However, compared with other materials, its triboelectric performance is poor.

  1. There are several modes of fiber/yarn TENG. The authors have summarized them but did not make enough discussion on the selection of modes. It is suggested to add more discussion on this.

A: Thank you for your suggestion. This article mainly discusses the structure and working principle of textile-teng, and most of the relevant research currently use CS mode or SE mode. Therefore, these two working methods are mainly introduced in 1.1section. There is a lot of literature on the working modes of TENG, so it is not discussed in detail here. A detailed description of the four approaches can be found in the literature cited by us in the article and figure 1b.

  1. In the application part of energy harvesting, there is no discussion on the outputs of these fiber/yarn TENG. The authors are suggested to add some discussion on the electric outputs.

A: Thank you for your suggestion. We have added a table to summarize it.

Table 1. Output of fiber/yarn based TENG

Refs

Work mode

Power density

Triboelectric material

[79]

CS

26 W m-3

PDMS/Ag-coated yarn

[81]

SE

336.2 μW/m

PAN/PVDF nanofibers/Acrylic

[82]

SE

60 mW m2

Spandex/Polyester

[85]

SE

8.43 mW/m2

Skin/PDMS

[89]

CS

12.5 μW/m

Polyacrylonitrile/PDMS

[91]

SE

1.25 W/m2

Nylon/PVDF

[92]

CS

3.5 μW/m2

Silk/PTFE

[93]

CS

93 mW/m2

PA66/PVDF-TrFE/Rubber

[98]

SE

245 μW/m

PU/PTFE

  1. More discussions on the challenge and potential solutions of fiber/yarn TENG for real applications are added in the section on challenges and summary.  

 A: Thank you for your suggestion, we have added more discussion in summary.

For fiber/yarn TENG, simple and reliable structure, flexible design, and size selection are its most significant advantages. According to the triboelectric series, the material selection is relatively fixed, mainly PDMS, PTFE, nylon, and other materials. These materials have their own advantages and disadvantages. For example, PDMS and fluoropolymers are extremely triboelectric, but poor in terms of mechanical properties and comfort. Textile materials such as nylon are excellent in comfort and processability, but most of them are tribo-positive materials with poor triboelectric performance. Therefore, there are often trade-offs in the selection of materials. The application of new technologies such as electrospinning of fluoropolymers, which are produced in the form of nanofibers and applied in fiber/yarn TENG. These methods greatly improve the wearing comfort of polymers. The addition of conductive fillers can further improve the triboelectric properties of these materials [105]. For wearable applications, there are fewer choices of existing materials. The synthesis and application of new polymers are expected.

Through structural design and different processing methods, fiber/yarn-TENG can possess the advantages of textiles, such as high elasticity, breathability and durability, and be used in fields such as energy harvesting, sensor design, and human-computer interaction. It is the popular directions of TENG at present. However, the contact surface area's size limited its energy harvesting application. The current common method is fabrication, converting the yarn into cloth. For use as a sensor, the susceptible inherent of triboelectricity is a hindrance. According to the triboelectric model, the triboelectric output depends on the exchange of electrons when the material is rubbed/contacted. This process depends on the external environment, such as humidity, temperature, and other influences. The change of environment will change the triboelectric performance, resulting in instability as sensor. Meanwhile, the accumulation of surface charges on triboelectric materials is the key to triboelectric output. The generation of triboelectric charges is often accompanied by rapid loss, so it is easy to change with time. A possible solution is to introduce special microstructures on the surface of triboelectric materials, such as superhydrophobic structures, to increase the stability of the triboelectric properties of materials and avoid the influence of environmental factors. Sealing of triboelectric devices is also a possible solution.

  1. On page 3, lines 107-109, the authors state that “temperature, humidity, sweating, cleaning, friction, and other conditions can have an impact on how well traditional electronic components work. Consequently, it is important to produce textiles that are self-sufficient.”. The authors are suggested to make clearer discussion.

A: Thank you for your suggestion. We have made corresponded change.

In addition, temperature, humidity, sweating, cleaning, friction, and other conditions can have an impact on how well traditional electronic components work [52]. Consequently, these issues have become major challenges in the design of smart textiles.

Reviewer 2 Report

Authors have detailed the development of fiber/yarn-based TENGs in recent years. The article is well written and has a clear structure. I suggest that it can be accepted in present form. Congratulations!

Author Response

Thank you very much for your attention and recommend for publication on our paper. 

Reviewer 3 Report

This review discusses about the fabrication, structure and application of Fiber/yarn-based TENG. The discussion is well summarized. Also, the key features responsible for the device performance have been addressed well. However, a minor revision is needed before publishing this manuscript in the Sensors journal as follows.

1. Please elaborate about Fig. 1a, nowhere is mentioned/described about Fig.1a.

2. Copyright should be mentioned at each figure caption.

3. It is advised to separate the sub-figures in each figure corresponding to each reference. In some figures there are partial separation such as in fig. 6, 6b is separated among three subfigures. In fig. 8, 8b & 8d are separated only. In fig. 9, 9a only separated. So please separate each sub-figure corresponding to reference.

4. Please add a reference which reported highest electrical output in section 5.1.

Author Response

This review discusses about the fabrication, structure and application of Fiber/yarn-based TENG. The discussion is well summarized. Also, the key features responsible for the device performance have been addressed well. However, a minor revision is needed before publishing this manuscript in the Sensors journal as follows.

  1. Please elaborate about Fig. 1a, nowhere is mentioned/described about Fig.1a.

A: Thank you for your suggestion, we discussed it in section 1.1 but forget to add refer of figure. Here is the discussion:

The model of the triboelectric principle is summarized by Daniel et al [31]. Based on this research Weon-Guk et al. considered that the charge transfer mechanisms could be categorized into three possible species which are responsible for triboelectricity: electrons, ions, and cleaved bulk materials (Fig.1a) [27]. In the electron transfer model, as two materials come into contact, charges are transferred according to the different charge affinities of the materials [32,33]. Despite the lack of conclusive experimental evidence, this assumption is widely accepted. Moreover, some research proved that transferred electrons are the primary species in contact electrification. The ion transfer model is normally applied in ionic polymers. In this model, mobile ions could transfer from one material to another when the materials are in contact [34,35]. The third model is material Transfer Model [36]. Through Kelvin probe force microscopy, CRS, and XPS analysis, Baytekin et al. found that the contact between the polymers is charged to produce a nano mosaic charge pattern, which is generated by the corresponding material transfer. The nanomaterial fragments transfer the electric charge, creating a triboelectric phenomenon [30].

  1. Copyright should be mentioned at each figure caption.

A: Thank you for your suggestion, we have made corresponded change.

  1. It is advised to separate the sub-figures in each figure corresponding to each reference. In some figures there are partial separation such as in fig. 6, 6b is separated among three subfigures. In fig. 8, 8b & 8d are separated only. In fig. 9, 9a only separated. So please separate each sub-figure corresponding to reference.

A: Thank you for your suggestion, we have made corresponded change.

  1. Please add a reference which reported highest electrical output in section 5.1

A: Thank you for your suggestion. We have added a table to summarize it.

Table 1. Output of fiber/yarn based TENG

Refs

Work mode

Power density

Triboelectric material

[79]

CS

26 W m-3

PDMS/Ag-coated yarn

[81]

SE

336.2 μW/m

PAN/PVDF nanofibers/Acrylic

[82]

SE

60 mW m2

Spandex/Polyester

[85]

SE

8.43 mW/m2

Skin/PDMS

[89]

CS

12.5 μW/m

Polyacrylonitrile/PDMS

[91]

SE

1.25 W/m2

Nylon/PVDF

[92]

CS

3.5 μW/m2

Silk/PTFE

[93]

CS

93 mW/m2

PA66/PVDF-TrFE/Rubber

[98]

SE

245 μW/m

PU/PTFE